# Ace-TN: GPU-Accelerated Corner-Transfer-Matrix Renormalization of Infinite Projected Entangled-Pair States

Addison D. S. Richards $\bullet$[1]⋆ and Erik S. Sørensen $\bullet$[1]†

1 Department of Physics and Astronomy, McMaster University, Hamilton, Ontario L8S 4M1, Canada

⋆ richaa12@mcmaster.ca ,   † sorensen@mcmaster.ca

## Abstract

The infinite projected entangled-pair state (iPEPS) ansatz is a powerful tensor-network approximation of an infinite two-dimensional quantum many-body state. Tensor-based calculations are particularly well-suited to utilize the high parallel efficiency of modern GPUs. We present Ace-TN, a modular and easily extendable open-source library developed to address the current need for an iPEPS framework focused on GPU acceleration. We demonstrate the advantage of using GPUs for the core iPEPS simulation methods and present a simple parallelization scheme for efficient multi-GPU execution. The latest distribution of Ace-TN can be obtained at https://github.com/ace-tn/ace-tn.

# 1  Introduction

Projected entangled-pair states (PEPS) approximate the quantum many-body state through a tensor-network representation of the state coefficient [1]. This representation allows for the simulation of systems with lattice sizes much greater than is accessible by exact techniques, and does not suffer from the sign problem of quantum Monte-Carlo. The PEPS ansatz can be further extended to the thermodynamic limit, referred to as the infinite-PEPS (iPEPS) ansatz. An iPEPS approximates an infinite system through iterative renormalization of boundary tensors surrounding a finite unit-cell of tensors assigned to the sites of a lattice [2]. This property makes iPEPS a useful tool for studying quantum many-body systems relevant to condensed matter. An important technique that has seen significant success in the calculation of an iPEPS [3], is the corner-transfer-matrix renormalization group (CTMRG) method [4–12], which we focus on in this work. Techniques for simulating PEPS are well-developed, and many excellent reviews are available [13–21].

Significant development effort has already been undertaken to build tensor-network software libraries for simulating quantum many-body systems. ExaTN [22], Quimb [23], and Cytnx [24–26] are general-purpose frameworks providing methods for efficient tensor-network contraction as well as some common algorithms encountered in quantum simulations. Some specialized libraries have been developed focusing on matrix-product states, such as TeNPy [27–29] and iTensor [30, 31]. While several of the already mentioned libraries are capable of simulating finite PEPS, Koala [32] was additionally developed to efficiently scale finite PEPS calculations on distributed systems. TeNeS [33, 34] implements CTMRG-based iPEPS simulations, taking advantage of OpenMP and MPI to parallelize large-scale calculations. The automatic differentiation capabilities of machine-learning frameworks, typically used to train neural networks, have been adapted to accurately and efficiently calculate iPEPS [35–38], and some libraries such as VariPEPS [39, 40] and peps-torch [41] have been developed to take advantage of this. The QuantumKitHub [42] contains many open-source packages for simulating quantum states with tensor networks, including PEPS [43]. Many libraries have also been developed to exploit physical symmetries in tensor contractions [44–49], which may be integrated with other frameworks. Each of these works have provided a significant

advancement towards efficient simulation of quantum systems. However, the application of graphics-processing unit (GPU) acceleration to typical iPEPS calculations based on CTMRG and imaginary-time evolution has not been fully exploited or analyzed.

The computational bottleneck of an iPEPS calculation is the calculation of projectors used to renormalize boundary tensors in the CTMRG algorithm. This step scales sharply as $O(\chi^3 D^6)$, where $\chi$ and $D$ are the bond dimensions of the iPEPS tensors with $\chi \propto D^2$. Given that the size of the bond dimension controls the degree of entanglement captured by the iPEPS, with $D = 1$ corresponding to a product state, it is often desirable to achieve the highest possible values of $(D, \chi)$ to accurately describe quantum phases and phase transitions. Fortunately, this cost arises from one or more large tensor contractions and is therefore amenable to speed-up by utilizing the high theoretical performance of modern GPUs. Despite this clear advantage, GPU-based implementations of CTMRG have not been widely adopted, and the available speed-up has not been previously reported. Our work therefore serves to both benchmark the application of GPU acceleration in CTMRG for both single- and multi-GPU systems, and to provide a simple open-source implementation accessible to the physics research community.

To this end, we have developed `Ace-TN`, an iPEPS simulation framework that is designed for GPU-acceleration from the ground up. `Ace-TN` is based entirely on a PyTorch backend to simplify CUDA integration, while the flexibility of PyTorch allows for `Ace-TN` to run on multi-core CPU systems as well. We have designed `Ace-TN` as a modular, high-level library with a pythonic style so that researchers may quickly adapt and extend the core iPEPS algorithms to their use case. As new iPEPS simulation techniques are developed, `Ace-TN` may be used to benchmark GPU or multithreaded CPU performance, as we demonstrate for standard ground-state calculation methods in Section 3. The structure of `Ace-TN` was designed to follow closely the notation that we present in Section 2, which is largely a review of the well-established iPEPS techniques. In Section 4, we describe how to build an iPEPS with `Ace-TN` and perform a complete ground-state calculation. We show how `Ace-TN` can be used to simplify the construction of custom model Hamiltonians with a large degree of flexibility, and we provide several examples of how to use `Ace-TN` to study models of current research interest.

## 2 Background

There exists several review articles discussing iPEPS and the CTMRG algorithm in detail [13–21]. In this section, we briefly summarize the main computational steps for clarity and to establish our notation.

### 2.1 iPEPS

The PEPS ansatz approximates a quantum state through tensor-network factorization of the state coefficient

$$
\sum_{s_1,s_2,\ldots,s_N} \Psi_{s_1,s_2,\ldots,s_N} |s_1, s_2, \ldots, s_N\rangle \approx \sum_{s_1,s_2,\ldots,s_N} F\left(A^{[s_1]}_{l_1 u_1 r_1 d_1}, A^{[s_2]}_{l_2 u_2 r_2 d_2}, \ldots, A^{[s_N]}_{l_N u_N r_N d_N}\right) |s_1, s_2, \ldots, s_N\rangle
$$

(1)

where $A^{[s_i]}_{l_i u_i d_i r_i}$ are rank-5 tensors which we refer to as site tensors. The site tensors are assigned the physical index $s_i$ of dimension $d$ and bond indices, $l_i, u_i, d_i, r_i$ each of dimension $D$, and $F$ is a function that defines a contraction between all pairs of connected bond indices. Each site tensor, labelled by index $i$, is associated with one unique site, $(x_i, y_i)$, of the square lattice.

The starting point of the iPEPS ansatz is the double-layer tensor network formed by contracting the tensor-network state coefficient in Eq. (1) with its conjugate. For illustrative pur-

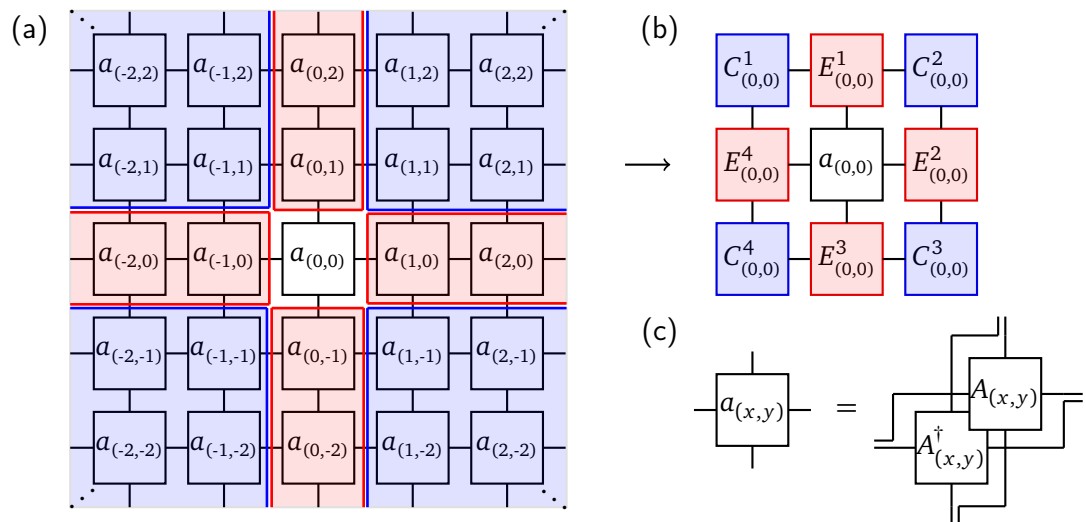

Figure 1: (a) Depiction of an infinite double-layer tensor network with coordinates $(x, y)$ defined relative to the central site $(x, y) = (0, 0)$. (b) Approximation of the infinite tensor network by boundary tensors $C_{(0,0)}^k$ and $E_{(0,0)}^k$ for $k = 1, 2, 3, 4$. (c) Contraction and bond fusion of site tensors, $A_{(x,y)}$, to form the $a_{(x,y)}$ tensors.

poses, it is useful to define the contracted and reshaped tensor

$$a_{(l_i l_i')(u_i u_i')(d_i d_i')(r_i r_i')} = \sum_{s_i} A_{l_i u_i d_i r_i}^{[s_i]} A_{l_i' u_i' d_i' r_i'}^{[s_i]\dagger} \tag{2}$$

with fused dimensions $D^2 \times D^2 \times D^2 \times D^2$ (see Fig. 1(c)). The bond indices $l_i$, $u_i$, $d_i$, and $r_i$ are represented as lines extending from the site tensors in the left, up, down, and right directions respectively in Fig. 1. We refer to the contraction of all $a$ tensors, as depicted in Fig 1(a) for an infinite system, as a double-layer tensor network. It is well known that the PEPS ansatz can be formulated for an infinite periodic system by suitable construction of boundary tensors, $C_i^k$ and $E_i^k$, which we refer to as corner-transfer matrices and edge tensors respectively, with $k = 1, 2, 3, 4$ assigned to each unique boundary tensor, as shown in Fig.1(b). We assume that the tensors form a periodic arrangement such that, for some unit-cell size $(N_x, N_y)$, $a_{(x,y)} = a_{(x',y')}$ for all $(x, y)$ with $x' = x \mod N_x$ and $y' = y \mod N_y$. In total, each site, $(x, y)$, in the unit cell is associated with 9 tensors: $A_{(x,y)}$, $C_{(x,y)}^k$, and $E_{(x,y)}^k$, for each $k = 1, 2, 3, 4$ as shown in Fig. 1(b). Assuming that each site in the system is unique, $9 \times N_x \times N_y$ tensors are therefore used to approximate the infinite system. The total number of tensors can be reduced by enforcing spatial symmetries beyond the already assumed translational symmetry of the unit cell.

If the site tensors and boundary tensors are determined to form a good approximation of the ground-state iPEPS, quantities depending on contraction of the infinite tensor network can then be calculated efficiently. For example, expectation values of local observables may be calculated by only contracting the local operator with a small number of site tensors and the relevant boundary tensors.

## 2.2 CTMRG

### 2.2.1 Directional Moves

The calculation of boundary tensors, $C_{(x,y)}^k$ and $E_{(x,y)}^k$, approximating an infinite extension of the $a_{(x,y)}$ tensors as depicted in Fig. 1, is achieved through iterative insertion and absorption

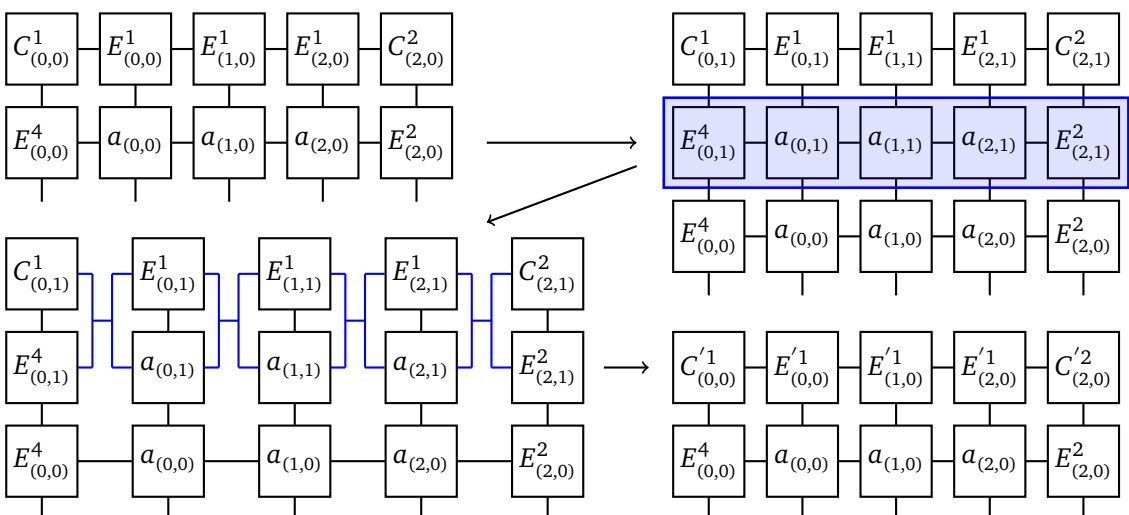

Figure 2: Steps involved in a single up-move for a system with $N_x = 3$ used to update the boundary tensors $C^1_{(x,y)}$, $E^1_{(x,y)}$, and $C^2_{(x,y)}$, for each position $(x, y)$ with $y = 0$ and $x = 0, 1, 2$. First, a $y = 1$ row of tensors is inserted above the $y = 0$ row. The $y = 1$ site tensors are then absorbed into the $y = 1$ boundary tensors forming updated $y = 0$ boundary tensors.

---

**Algorithm 1:** Up-move

**Input:** Row position, $y \in \{0, 1, \ldots N_y - 1\}$, and `ipeps` with $N_x \times N_y$ unit cell and containing tensors $C^k_{(x,y)}$, $E^k_{(x,y)}$, and $A_{(x,y)}$.

**Output:** Sets of updated tensors $C^{'1}$, $E^{'1}$, and $C^{'2}$.

**for** $x \leftarrow 0$ *to* $N_x - 1$ **do**
$\quad | \quad P^1_x, P^2_x \leftarrow$ Calculate projectors using `ipeps` tensors (see Algo. 2)
**end**

**for** $x \leftarrow 0$ *to* $N_x - 1$ **do**
$\quad \Big| \quad C^{'1}_{(x,y)} \leftarrow \texttt{contract}\Big(E^4_{(x,y+1)}, C^1_{(x,y+1)}, P^1_x\Big)$
$\quad \Big| \quad C^{'2}_{(x,y)} \leftarrow \texttt{contract}\Big(E^2_{(x,y+1)}, C^2_{(x,y+1)}, P^2_{x-1}\Big)$
$\quad \Big| \quad E^{'1}_{(x,y)} \leftarrow \texttt{contract}\Big(P^2_x, E^1_{(x,y+1)}, A_{(x,y+1)}, A^\dagger_{(x,y+1)}, P^1_{x-1}\Big)$
**end**

**return** $C^{'1}$, $E^{'1}$, $C^{'2}$

---

of rows or columns of the double-layer tensor network. In Fig. 2, we show how a row of tensors is inserted and absorbed into the boundary tensors. Each absorption step is referred to as a directional move. The direction associated with a directional move indicates the relative location of the inserted row or column of tensors. When necessary, the direction of a particular move is stated explicitly. For example, Fig. 2 shows a simplified depiction of an *up-move* relative to the row of tensors at $y = 0$. The exact steps involved in an up-move, including the update of some boundary tensors that are not shown in Fig. 2, are given in Algorithm 1. In an up-move, the $y = 1$ row of tensors are replicated and inserted above the $y = 0$ row. Every $C^1_{(x,y)}$, $C^2_{(x,y)}$, and $E^1_{(x,y)}$ tensor associated with $y = 0$ is then updated by absorbing the $y = 1$ tensors into the $y = 1$ boundary tensors. The *left-move*, *right-move*, and *down-move* are defined similarly. For a unit cell with length $N_y$, the entire unit cell is absorbed from the upwards direction by performing and up-move starting with row $y = N_y - 1$, and using the update $y = N_y - 1$

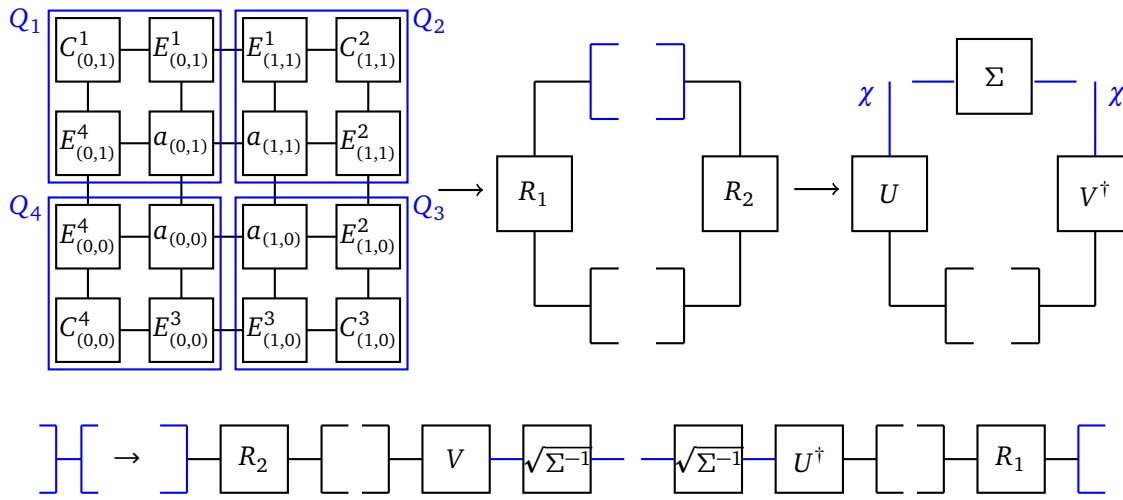

Figure 3: Construction of projectors used in boundary-tensor update for a $y = 0$ up-move. First, quarter tensors, $Q_k$ for $k = 1, 2, 3, 4$, are constructed by contracting the relevant tensor. The quarter tensors are contracted to form $R_1$ and $R_2$ tensors with $O(D^{12})$ computational cost. The $R_1$ and $R_2$ tensors are contracted along the bonds to be projected (highlighted in blue) and decomposed to obtain singular values and vectors of the full tensor network. The singular value spectrum is truncated, retaining the the highest $\chi$ values and corresponding singular vectors. The resulting tensors are then used to form projectors approximating a decomposition of the identity tensor which we used to depict tensor absorption in Fig. 2.

---

**Algorithm 2:** Calculate projectors (full-system)

**Input:** Site positions used to build quarter tensors, `sites`, direction, $k$, boundary bond dimension, $\chi$, and `ipeps` containing tensors $C_{s_i}^k$, $E_{s_i}^k$, and $A_{s_i}$.

**Output:** Projectors $P_1$, $P_2$.

$Q_1, Q_2, Q_3, Q_4 \leftarrow$ Build quarter tensors given `ipeps`, `sites`, and $k$

$R_1 \leftarrow \texttt{contract}(Q_1, Q_4)$

$R_2 \leftarrow \texttt{contract}(Q_2, Q_3)$

$U, \Sigma, V \leftarrow \texttt{SVD}(\texttt{contract}(R_1, R_2))$

Truncate dimensions of $U$, $\Sigma$, and $V$, to new value $\chi$

$P_1 \leftarrow \texttt{contract}(R_1, U^\dagger, \sqrt{\Sigma^{-1}})$

$P_2 \leftarrow \texttt{contract}(R_2, V, \sqrt{\Sigma^{-1}})$

**return** $P_1$, $P_2$

---

boundary tensors to perform an up-move for row $y = N_y - 2$, and so on until completing the final up-move for row $y = 0$. The boundary tensors are considered to be converged when they reach a fixed point with respect to the absorption of a unit cell.

### 2.2.2 Projector Calculation

Absorptions clearly cannot always be executed exactly, since each absorption step increases the boundary bond dimension of the updated boundary tensors by a factor of $D^2$. In practice, a maximum boundary bond dimension $\chi^{\max}$ is specified such that if at CTMRG iteration $i$, the updated boundary bond dimension $D^2 \chi^{(i-1)}$ exceeds $\chi^{\max}$, which will typically occur after a few directional moves, the updated boundary tensors must be projected onto a lower-dimensional subspace of dimension $\chi^{(i)} \leq \chi^{\max}$ in order to proceed with further absorptions.

The relevant subspace projectors are determined by using the truncated singular values and singular vectors of the contracted double-layer tensor network, as shown in Fig. 3.

Each projector calculation involves the formation of *quarter* tensors, which we denote by $Q_k$ for $k = 1, 2, 3, 4$ as shown in Fig. 3. The projectors used in the absorption step are typically determined using a truncated decomposition of a tensor obtained from a contraction of some combination of quarter tensors. In the original work of Ref. [3], the authors used a combination of eigendecompositions of the symmetric tensors obtained by contraction of individual quarter tensors with themselves to obtain projectors. A modified approach was later suggested in Ref. [50] such that projectors are calculated using the full double-layer tensor network. We refer to projectors obtained in this way as *full-system* projectors. The steps for computing full-system projectors are provided in Algorithm 2 and depicted in Fig. 3. Alternatively, it was also suggested in Ref. [50] that projectors may be formed by a contraction of only half of the tensor network. We refer to these projectors as *half-system* projectors. This approach avoids two large contractions of leading order $O(D^{12})$, reducing the required contraction FLOPS to approximately 1/3 that of the full-system method, at the cost of disregarding correlations within half of the system.

## 2.3   Tensor Update

We consider the standard iterative update procedure [1] based on imaginary-time evolution,

$$|\Psi_0\rangle = \lim_{\beta \to \infty} \frac{e^{-H\beta}|\Psi_t\rangle}{\|e^{-H\beta}|\Psi_t\rangle\|} \tag{3}$$

to determine ground-state iPEPS $|\Psi_0\rangle$ from trial state $|\Psi_t\rangle$ given the Hamiltonian $H$. The imaginary-time-evolution operator is decomposed into a product of two-site gates $g_{ij} = e^{-h_{ij}\Delta\tau}$ by Trotter-Suzuki factorization [51] for sufficiently small $\Delta\tau$ and local two-site Hamiltonian terms $h_{ij}$. The initial state is then projected towards the ground-state tensor-network representation by repeatedly applying the two-site gates. Given that this operator is local to a single bond, it is beneficial to perform preliminary decompositions of the site tensors connected by the bond [52]. We use QR decompositions such that $A_i = A_i^Q a_i^R$ and $A_j = A_j^Q a_j^R$ as depicted in Fig. 4, where $a_i^R$ and $a_j^R$ are each reduced tensors of shape $(dD, D, d)$. The gate is then applied to the reduced tensors. Updated reduced tensors with the same shape as the original reduced tensors must be determined to approximate the product of $a_i^R$, $a_j^R$, and $g_{ij}$. This is most accurately achieved by minimizing the distance,

$$d(\tilde{a}_i^R, \tilde{a}_j^R) = \||\psi_{\tilde{a}_i^R, \tilde{a}_j^R}\rangle - |\psi'\rangle\|^2 = \langle\psi_{\tilde{a}_i^R, \tilde{a}_j^R}|\psi_{\tilde{a}_i^R, \tilde{a}_j^R}\rangle - \langle\psi_{\tilde{a}_i^R, \tilde{a}_j^R}|\psi'\rangle - \langle\psi'|\psi_{\tilde{a}_i^R, \tilde{a}_j^R}\rangle + \langle\psi'|\psi'\rangle \tag{4}$$

between the state $|\psi'\rangle$ obtained from the reduced tensors evolved by the two-site gate, and the state $|\psi_{\tilde{a}_i^R, \tilde{a}_j^R}\rangle$ obtained from the approximate tensors, $\tilde{a}_i^R$ and $\tilde{a}_j^R$, to be determined in the tensor optimization step. This distance is typically minimized by an alternating least-squares (ALS) method [1], yielding the approximate tensors. The ALS routine is performed by repeatedly minimizing the distance function for one tensor while keeping the other fixed, alternating the choice of fixed tensor until convergence. For example, by fixing the tensor $\tilde{a}_j^R$, the distance as defined in Eq. (4) for the variable $\tilde{a}_i^R$ has the form

$$d(\tilde{a}_i^R, \tilde{a}_i^{\dagger R}) = \tilde{a}_i^{\dagger R} R_i \tilde{a}_i^R - \tilde{a}_i^{\dagger R} S_i - S_i \tilde{a}_i^R + C \tag{5}$$

where $R_i$ and $S_i$ are the tensors as depicted in Fig 4 and $C$ is a constant. The approximated tensor $\tilde{a}_i^R$ is then determined by solving $R_i \tilde{a}_i^R = S_i$. Given the updated $\tilde{a}_i^R$, an updated estimate for $\tilde{a}_j^R$ is similarly obtained using fixed $\tilde{a}_i^R$ by constructing $R_j$ and $S_j$ tensors and solving

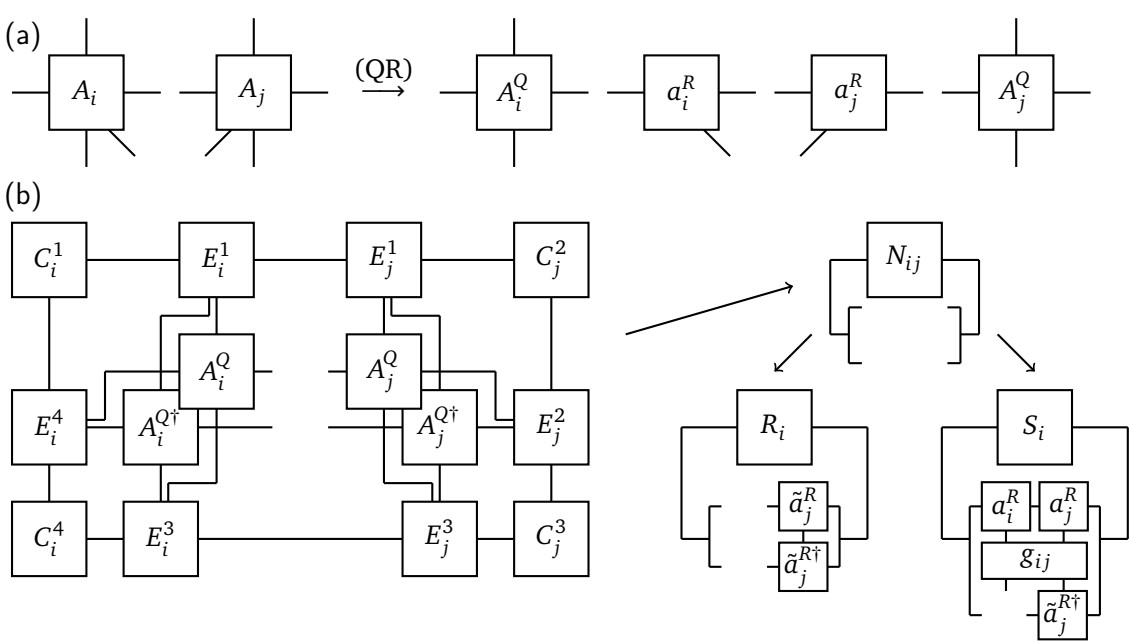

Figure 4: Construction of tensors used in the full update of $A_i$ and $A_j$. (a) $A_i$ and $A_j$ are QR-decomposed yeilding the $A_i^Q$, $a_i^R$, $A_j^Q$, and $a_j^R$ tensors. (b) $A_i^Q$ and $A_j^Q$ are used to build the norm tensor, $N_{ij}$. The $R_i$ and $S_i$ tensors are constructed from $N_{ij}$ an used to determine the approximated tensor $\tilde{a}_i^R$ in a single ALS step.

---

**Algorithm 3:** Full Update

---

**Input:** Site tensors, $A_i$ and $A_j$, sets of boundary tensors, $C_i$, $E_i$, $C_j$, and $E_j$, and two-site gate $g_{ij}$.

**Output:** Updated site tensors, $A_i'$ and $A_j'$.

$A_i^Q, a_i^R, A_j^Q, a_j^R \leftarrow$ QR-decompose $A_i$ and $A_j$.

$N_{ij} \leftarrow$ Build norm tensor using $A_i^Q$, $C_i$, $E_i$, $A_j^Q$, $C_j$, and $E_j$.

$\tilde{a}_i^R, \tilde{a}_j^R \leftarrow$ Initialize using truncated SVD of contracted $a_i^R$, $a_j^R$, and $g_{ij}$.

**while** ALS not converged **do**

> $R_i, S_i \leftarrow$ Build $R_i$ and $S_i$ tensors using $\tilde{a}_j^R$
>
> $L_i \leftarrow$ Cholesky-decompose $R_i + \epsilon \mathbb{1}$
>
> $\tilde{a}_i^R \leftarrow$ Triangular-solve $L_i L_i^\dagger \tilde{a}_i^R = S_i$
>
> $R_j, S_j \leftarrow$ Build $R_j$ and $S_j$ tensors using $\tilde{a}_i^R$
>
> $L_j \leftarrow$ Cholesky-decompose $R_j + \epsilon \mathbb{1}$
>
> $\tilde{a}_j^R \leftarrow$ Triangular-solve $L_j L_j^\dagger \tilde{a}_j^R = S_j$

**end**

$A_i', A_j' \leftarrow$ Recompose $A_i', A_j'$ using $A_i^Q, \tilde{a}_i^R, A_j^Q, \tilde{a}_j^R$

**return** $A_i', A_j'$

---

$R_j \tilde{a}_j^R = S_j$. After sufficient iterations of ALS, the updated site tensors are obtained from the updated reduced tensors as $A_i' = A_i^Q \tilde{a}_i^R$ and $A_j' = A_j^Q \tilde{a}_j^R$. This defines the full update.

The linear system, $R_i \tilde{a}_i^R = S_i$, solved in each ALS iteration, is generally poorly conditioned. This problem may be alleviated to an extent by positive approximation of the norm tensor and gauge fixing [53]. Additionally, we find it beneficial to add a small regularization factor, $\epsilon$, to the diagonal of the $R$ matrix, $R \to R + \epsilon \mathbb{1}$. This improves conditioning and enforces $R$ to

be positive-definite. Since $R$ is already Hermitian, we use the Cholesky decomposition of $R$ to efficiently solve the linear system in each ALS step. The conditioning of the ALS typically becomes worse as the bond dimension is increased. In some cases, it may be necessary to adjust the regularization factor to a larger value. Although, we use $\epsilon = 10^{-12}$ for all results unless otherwise stated. The cost of the ALS sweep scales only as $O(d^3 D^6)$, which is quickly out-scaled by the $O(D^{12})$ projector calculation as $D$ is increased. The exact steps that we use to perform a full update are given in Algorithm 3.

After convergence of ALS, the new tensors may then be reproduced throughout the infinite lattice using CTMRG in preparation for the next tensor update. However, it has been shown that a full renormalization of the boundary tensors is not required after every full update [53]. It is enough to absorb the updated tensors into the boundary tensors connected by the updated bond and proceed with the next full update. This combination of full update and boundary renormalization steps is referred to as a fast-full update [53]. The fast-full update is a favourable modification of the standard update procedure, since it greatly reduces the number of boundary renormalization steps required to reach convergence.

Alternatively to the fast-full update, a gradient-based tensor update has also been developed [54]. This type of update has been shown to obtain a more precise ground state and converge faster than the standard full-update in some cases [54]. Gradient computation may be simplified by utilizing automatic differentiation (AD) techniques ubiquitous in machine-learning frameworks [35–38]. However, these methods do not reduce the leading computational cost, as the converged boundary tensors must be obtained after every gradient step. The backward pass of AD may suffer from stability issues and requires some additional management of compute or memory cost of intermediate tensor construction. Furthermore, the variational update requires reconvergence of the boundary tensors at every step, in contrast to the fast-full update which updates site tensors and boundary tensors simultaneously. A useful hybrid approach may be to use imaginary-time evolution to project the site tensors close to the ground state and further refine the result with an AD-based variational solver.

## 2.4 Measurement

Once the converged iPEPS tensors are obtained, local measurements are efficiently calculated by constructing reduced density matrices (RDM) from the converged tensors. In Fig. 5, we show an example of how the RDM, $\rho_{ij}$, is formed and used to calculate a measurement of an arbitrary two-site observable acting on bond sites $i$ and $j$, $\langle O_{ij}\rangle = \langle\Psi|O_{ij}|\Psi\rangle/\langle\Psi|\Psi\rangle$. Single-site or multi-site observables can also be measured through suitable RDM construction. The

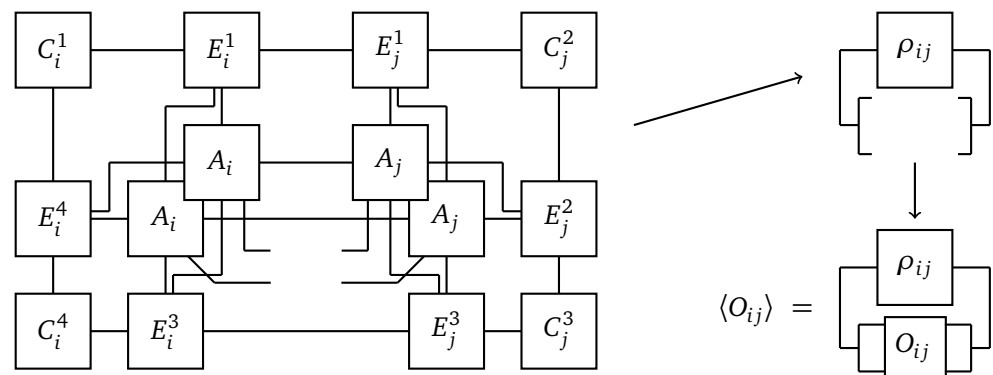

Figure 5: Example measurement of an arbitrary two-site operator $O_{ij}$ using iPEPS tensors. The RDM, $\rho_{ij}$, is constructed and contracted with $O_{ij}$ to calculate the expectation value $\langle O_{ij}\rangle$.

measurement stage may have high computational cost for multi-site or long-range observables. However, measurement is only necessary once, in post-processing the result of many imaginary-time evolution steps.

# 3 Performance

## 3.1 Technical Details

The CTMRG projector calculation, as discussed in Section. 2.2.2, is the main computational bottleneck, and therefore deserves the most attention. As previously mentioned, this part of the calculation is almost entirely dominated by a single SVD and tensor contractions of $O(D^{12})$ cost. Since at minimum $D^2$ singular values are discarded in every projector calculation, the $O(D^{12})$ scaling of the full SVD may be reduced to the more favourable scaling $O(D^{10})$ through the randomized SVD algorithm [55], which we refer to as RSVD. The RSVD uses a power method which is well suited for GPU acceleration [56]. Typically, only a small number of power iterations are required. We use 2 power iterations with target rank $\chi^{\max}$ and an oversampling factor of 2 in all presented results. Furthermore, the RSVD is most efficient when only a small fraction of the singular values are computed. In the case of CTMRG, it is fortunate that the fraction of computed singular values is approximately $\chi^{(i)}/D^2\chi^{(i-1)} \approx 1/D^2$, which becomes less than 0.01 in the often desired case of $D > 10$. Since errors introduced in the RSVD are controllable by adjusting power iterations, and the full set of singular values and vectors are never utilized except possibly in the first few iterations of boundary tensor initialization, it is always advantageous to use the RSVD over the full-rank SVD. In practical calculations, the singular values less than a certain threshold value are typically discarded, so that $\chi^{(i)} \leq \chi^{\max}$ at any point in the simulation. However, we always truncate $D^2\chi^{(i)}$ to $\chi^{\max}$ for benchmarking purposes. The minimum value of $\chi^{max}$ is typically taken to be $D^2$. We use this minimum value in our benchmark calculations since it provides the most balanced comparison of operations. Increasing $\chi^{\max} > D^2$ only increases tensor sizes, which pushes tensor contractions towards a more compute-bound regime, such that the GPUs are saturated for a smaller value of $D$. For simplicity and consistency with conventional notations, we refer to $\chi^{\max}$ as $\chi$.

Only a modest amount of memory is needed to store an entire iPEPS, even for the largest system sizes. The memory requirements needed to store each site tensor and the associated boundary tensors scales as $O(D^4)$, which amounts to less than 0.05GB for a $8 \times 8$ system consisting of $9 \times N_x \times N_y$=576 tensors with FP64 data type. This means that almost all the GPU memory may be utilized to construct the largest intermediate tensors of size $O(D^8)$. In a multi-GPU calculation, all of the iPEPS tensors can be simply replicated across each GPU. These rather modest memory requirements mean that most simulations can be accommodated on a typical GPU with 40 GB global memory, except for in the more extreme cases (e.g. $D \geq 12$). At this point, the memory consumption of the $O(D^8)$-size intermediates rises sharply as $D$ is increased further, and more advanced techniques such as exploiting symmetries, sharding tensors across multiple GPUs, or pipelining host-device transfers are necessary.

## 3.2 Single GPU

### 3.2.1 Projector Calculation

First and foremost, we benchmark the CTMRG projector calculation as described in Section 2.2.2, which is the main computational bottleneck. We have benchmarked the projector calculation using one NVIDIA A100 GPU with 19.5 TFLOPS/s theoretical FP64 tensor-core peak performance and one Intel Xeon Gold 6148 with 20 cores and maximum turbo frequency of 3.7 GHz

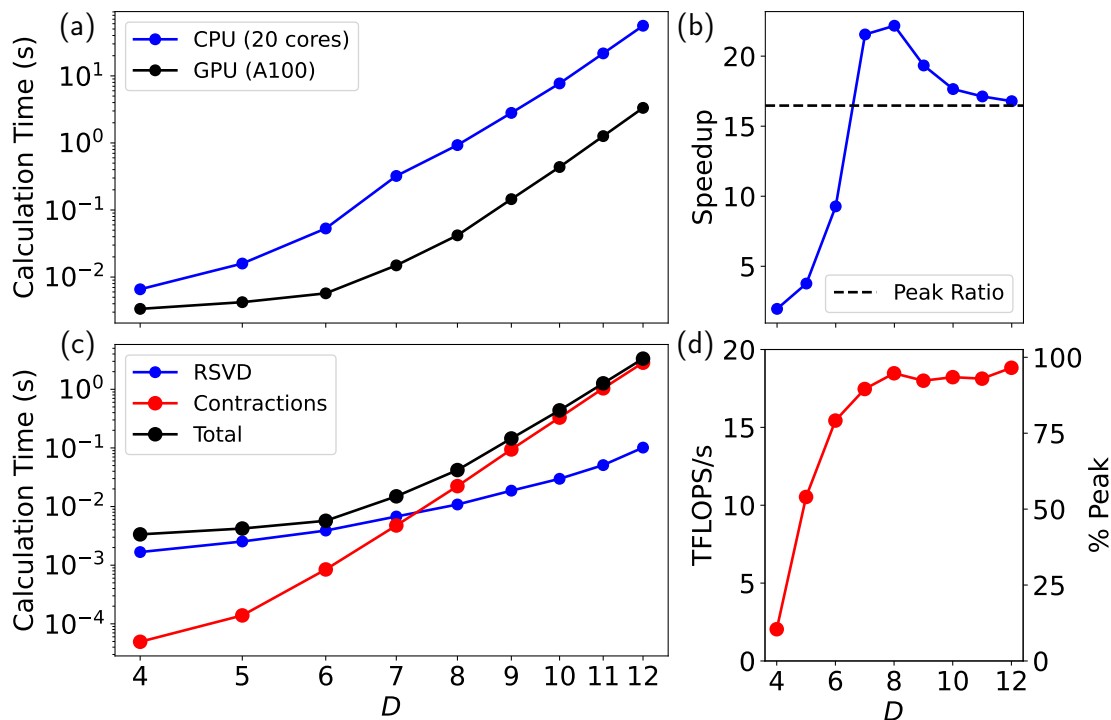

Figure 6: (a) Total runtime for a full-system projector calculation using a single A100 GPU and Xeon Gold 6148 CPU using all 20 cores with max turbo frequency of 3.7 GHz. (b) Speed-up achieved along with the peak performance ratio, using 19.5 TFLOPS/s for the GPU and 1.184 TFLOPS/s for the CPU. (c) Breakdown of the full-system projector calculation using one A100. (d) TFLOPS/s achieved for the $O(D^{12})$ contractions involved in the full-system projector calculation. Beyond $D = 8$, greater than 90% peak performance is achieved.

providing 1.184 TFLOPS/s FP64 peak performance. The total CPU and GPU runtime is plotted in Fig. 6(a), while the speed-up achieved is plotted in Fig. 6(b). We show that GPU execution is always faster than CPU execution, with GPU execution becoming more efficient for larger bond dimensions as tensor sizes become large enough to saturate the GPU. For the largest bond dimensions, the speed-up is approximately equal to the ratio of GPU and CPU peak performance. This is expected since this limit is dominated by the largest tensor contractions, which are efficiently executed on each system.

The projector calculation is dominated by two parts; the SVD and $O(D^{12})$ contractions. We break down the total runtime based on these parts in Fig. 6(c). As mentioned in Section 3.1, we always use the RSVD with favourable $O(D^{10})$ scaling over the full-rank SVD in the projector calculation. We find that tensor contractions are the primary contribution to the total runtime except in the case of small bond dimension ($D < 7$), at which the RSVD and contraction runtimes are comparable, as shown in Fig. 6(c). $D = 7$ roughly corresponds to the bond dimension at which tensor contractions can saturate the GPU, achieving close to 90% peak performance as shown in Fig. 6(d).

### 3.2.2 Tensor Update

Next, we measure the performance of the full update used in imaginary-time evolution. The primary contributions to full update execution are construction of the norm tensor as depicted in Fig. 4, and the alternating least-squares loop described in Algorithm 3. We consider 10 ALS

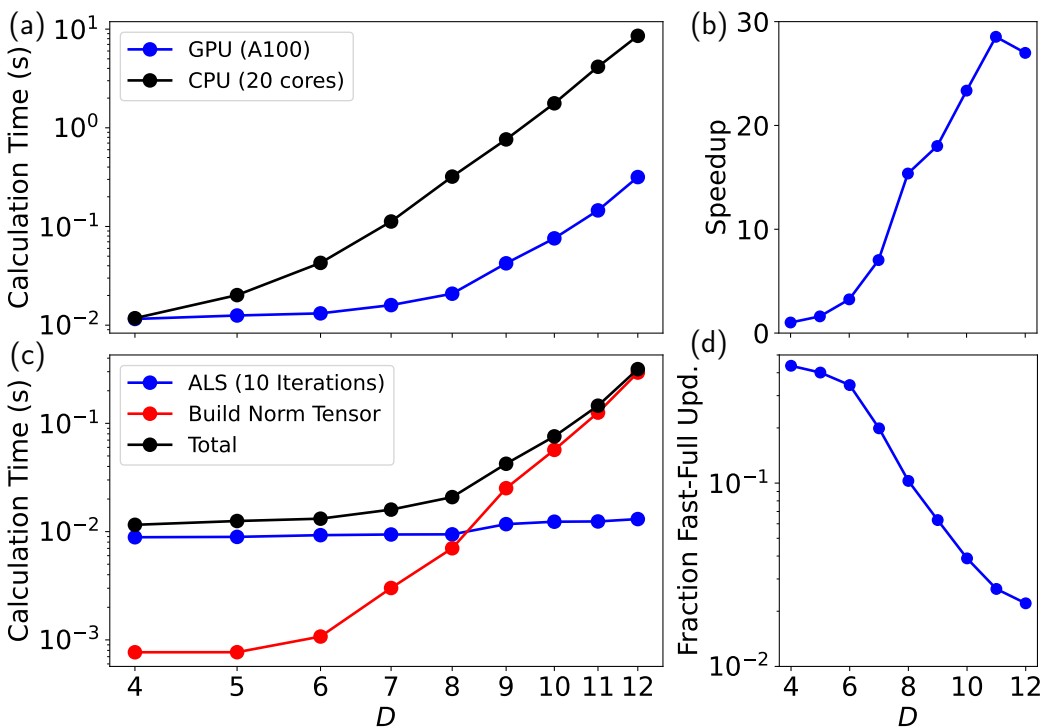

Figure 7: (a) Total runtime for a full update calculation using a single A100 GPU and Xeon Gold 6148 CPU using all 20 cores with max turbo frequency of 3.7 GHz. (b) Total speed-up achieved. (c) Breakdown of the full update calculation using one A100. (d) Fraction of runtime contributing to the fast-full update, consisting of one full update and four full-system projector calculations.

iterations for comparison purposes. We show the GPU and CPU runtimes for a full update in Fig 7(a) and the corresponding speed-up in Fig 7(b). We find that GPU execution of the full update is generally much more efficient than CPU execution, reaching a speed-up up factor of close to 30 for the largest bond dimensions considered. Breaking down the calculation runtime into ALS and norm tensor build components in Fig 7(c) shows that the runtime is dominated by the ALS loop in the low bond dimension regime and the norm tensor build for large bond dimensions ($D > 8$). We also show in Fig 7(d) the relative contribution of the full update runtime to the fast-full update runtime. The fast-full update that we consider consists of one full update followed by absorption of the updated tensors into the boundary tensors connected by the bond, as described in the original development of the fast-full update [53]. The full update is shown to contribute a non-negligible amount to the fast-full update, with decreasing contribution as $D$ increases. We also note that the norm tensor construction scales with $\chi$ as $O(\chi^2)$, while the CTMRG projector calculation scales as $O(\chi^3)$ implying that the contribution of the full update to the fast-full update will decrease as $\chi$ is increased.

## 3.3   Single-Node Multi-GPU

Each CTMRG directional move can be easily parallelized by dividing the required projector calculations (i.e. the first loop in Algorithm 1) across available GPUs. The resulting projectors are then gathered on all devices using the NVIDIA Collective Communications Library (NCCL) all-gather operation and used to update boundary tensors on each GPU. Since the up and down moves update boundary tensors independently and are always performed together, we combine these moves in our parallelization scheme and refer to the combined move as an up-

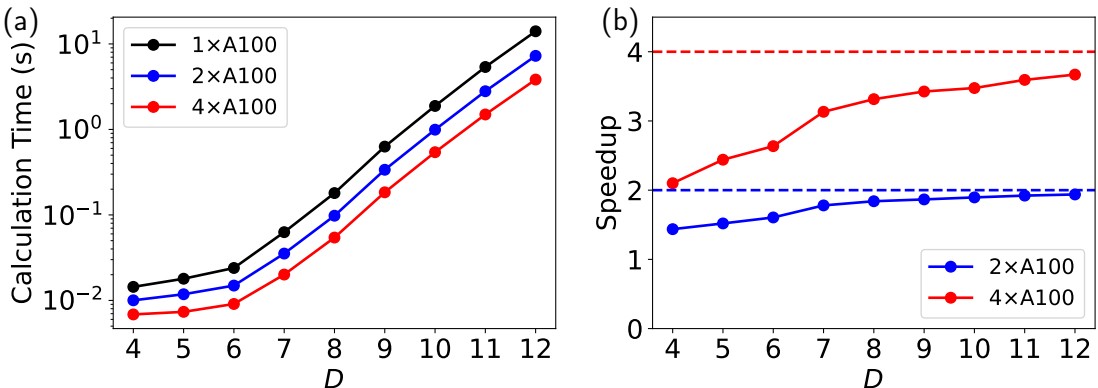

Figure 8: (a) Iteration time for an up-down move using multiple GPUs on a single node. (b) The corresponding Speedup achieved over single-GPU execution. Dashed lines in (b) indicate perfect parallel efficiency.

down move. Similarly, the left and right moves are combined and performed together. Fig. 8 shows the total runtime and speed-up achieved for an up-down move for a system with 2×2 unit cell using multiple GPUs on a single node with fully-connected NVLink interconnect. For a 2×2 unit cell, a single up-down move requires four projector calculations, and therefore our parallelization scheme is best implemented by using a number of GPUs that is a divisor of four in this case. We show that our multi-GPU calculations approach the optimal speed-up as $D$ is increased.

This simple parallelization scheme is also naturally incorporated into the fast-full update. In our multi-GPU implementation of the fast-full update, we perform the tensor update on a single GPU and broadcast the result to all other GPUs. Once the updated tensors are received, all GPUs are then used to independently calculate the projectors required to absorb the updated tensors. The tensor update therefore currently limits the fast-full update parallel efficiency to an extent, since we do not parallelize this step. However, the tensor update contributes only a small fraction of the fast-full update runtime, especially for large values of $D$, as shown in Figure 7 (d). Given the modest memory usage, we replicate all iPEPS tensors across each GPU so that only updated tensors and calculated projectors must be communicated between the GPUs.

## 4  Library Overview

### 4.1  Getting Started

Since our calculations primarily consist of tensor operations, we chose to develop `Ace-TN` using the high-level interfaces provided by PyTorch. This approach is advantageous, as we have built our library based on a simple pythonic programming style, allowing for quick adoption of our methodology or modification of our code. For example, the user may define custom Hamiltonians and operators to suit their model of interest with little effort.

#### 4.1.1  Installation and Usage

We provide a packaged distribution of `Ace-TN` on the public PyPI registry to enable easy installation. This can be achieved by:

```
1 pip install acetn
```

Alternatively, `Ace-TN` can be installed directly from GitHub using:

```
1  pip install git+https://github.com/ace-tn/ace-tn
```

`Ace-TN` is designed to be easily incorporated in common Python scripting workflows allowing researchers to easily set up complicated calculations and perform post-processing data analysis. We expect that the most common usage of `Ace-TN` in single-GPU and single-CPU calculations will be simply:

```
1  python script.py
```

with `script.py` including the steps required to setup and execute an iPEPS ground-state calculation. In addition, it may be useful to include steps such as computing measurements of observables, adjusting model Hamiltonian parameters, and saving the converged iPEPS tensors to disk for further post-processing or for initializing a separate calculation. We demonstrate how this can be be achieved in the following subsections, and we provide several example scripts in the `Ace-TN` source repository.

### 4.1.2 Multi-GPU Usage

To run a calculation using multiple GPUs, we use the torchrun utility provided by PyTorch to simplify the distributed setup. Running a calculation with multiple GPUs therefore does not require many changes beyond that of single-GPU execution. Any script involving our CTMRG and/or fast-full update implementations can take advantage of a multi-GPU environment with the command:

```
1  torchrun --nproc-per-node=N script.py
```

Provided $N$ GPUs are available, $N$ processes will be spawned and assigned to each GPU. We use the NVIDIA Collective Communications Library (NCCL) backend to initialize the process group and use NCCL collective operations as implemented in the distributed module of PyTorch for all operations involving GPU communication. Users interested in extending the multi-GPU capability of `Ace-TN`, for example, by implementing a multi-GPU tensor update may follow a similar approach. Although we have not attempted to scale the CTMRG calculation across multiple nodes, we expect that this may be achieved by using the multi-node capabilities of `torchrun` with little additional effort in cases where it is necessary to push the CTMRG calculation to the largest possible system sizes.

## 4.2 The iPEPS Class

### 4.2.1 Instantiating an iPEPS

The core functionality of `Ace-TN` is controlled by the `Ipeps` class. The user may initialize an `Ipeps` instance by providing a valid `config` dictionary to the constructor, specifying the system dimensions and unit-cell size as shown in Listing 1. Optionally, arguments may be specified controlling the choice of algorithms used in the boundary-tensor renormalization and tensor update methods, predefined model and model parameters, target device, and data type. By default, we use the GPU memory to store all tensors and perform all tensor computations using the GPU if a CUDA-capable device is available. `Ace-TN` may be executed on a CPU as well, using any linear algebra backend such as MKL or OpenBLAS accessible by `PyTorch` by setting the device to "cpu".

In Listing 1, we show how to initialize an iPEPS with 2×2 unit-cell, physical dimension 2, and bond dimensions $(D, \chi) = (4, 16)$. The AFM Heisenberg model,

$$H = J \sum_{\langle ij \rangle} \vec{S}_i \cdot \vec{S}_j \qquad (6)$$

is set as the model used for evolution of the iPEPS. We provide a preset implementation of the Heisenberg model, given that this is a typical model used in benchmarks and testing. Although, we expect users will benefit more generally by the flexibility of `Ace-TN` to support custom model specifications as described in Section 4.3.

```python
from acetn.ipeps import Ipeps

# Set options in ipeps_config
config = {
    "dtype": "float64"
    "device": "cuda"
    "TN":{
        "nx": 2,
        "ny": 2,
        "dims": {"phys": 2, "bond": 4, "chi": 16},
    },
    "model":{
        "name": "heisenberg",
        "params": {"J": 1.0},
    },
}
# Initialize an iPEPS
ipeps = Ipeps(config)
```

Listing 1: Example instantiation of the Ipeps class. An iPEPS is created with $2 \times 2$ unit cell in GPU memory using FP64 data type. The model is optionally set to the Heisenberg model for imaginary-time evolution and measurement.

### 4.2.2 Imaginary-Time Evolution and Measurement

In Listing 2, we show the basic workflow of an iPEPS calculation. The iPEPS is evolved closer to the ground state, starting with a small number of steps at the relatively large imaginary-time step value of $\Delta\tau = 0.1$, followed by many more steps with decreasing $\Delta\tau$. We find that a typical calculation may take close to 1000 imaginary-time steps to reach convergence. After evolution, measurements are calculated and the tensors are saved to `"ipeps.pt"`. The result can be loaded in a later calculation using `ipeps.load("ipeps.pt")` as needed.

```python
# Evolve the ipeps
ipeps.evolve(dtau=0.1, steps=10)
ipeps.evolve(dtau=0.01, steps=500)
ipeps.evolve(dtau=0.005, steps=500)
ipeps.evolve(dtau=0.001, steps=200)
# Calculate measurements
measurements = ipeps.measure()
# Save the evolved tensors to "ipeps.pt"
ipeps.save("ipeps.pt")
```

Listing 2: Example workflow for a typical iPEPS ground-state calculation. An iPEPS instance is created with 2×2 unit cell for the Heisenberg model. The iPEPS is then evolved in imaginary time and the energy is measured. The tensors are saved to ipeps.pt at the end of the script.

The results and calculation times obtained using the workflow in Listing 2 with an A100 GPU for a few moderately-sized bond dimensions are given in Table 1. We obtain energies and staggered magnetization values in agreement with Ref. [53] which should be compared with the quantum Monte Carlo estimates of -0.669437 and 0.30703 respectively [57]. The runtimes presented in Table 1 provide a representative sample of typical calculation runtimes. Assuming approximately 1000 ITE iterations are required to reach convergence, the iPEPS

calculations performed for these systems are expected to be completed within 5-20 minutes. As bond dimension increases, it can be seen in Table 1 that the CTM update begins to dominate the runtime. The large bond-dimension regime therefore benefits the most from our multi-GPU parallelization implementation, while calculations involving only a small bond dimension may be completed quickly enough on a single GPU as to not necessitate the use of a multi-GPU node.

| $D$ | Energy | Stag. Mag. | Time/Iter.(s) | Full Upd.(s) | CTM Upd.(s) |
|---|---|---|---|---|---|
| 6 | -0.669209 | 0.339970 | 0.278361 | 0.097723 | 0.179485 |
| 7 | -0.669266 | 0.335239 | 0.489497 | 0.120120 | 0.368218 |
| 8 | -0.669306 | 0.331353 | 1.011134 | 0.156182 | 0.853802 |

Table 1: Results for Heisenberg model using a single A100 GPU. Half-system projectors were used with RSVD (2 power iterations) using the script shown in Listing 2. One iteration consists of a fast-full update applied to all unique bonds.

### 4.2.3 Accessing Tensors

For development and post-processing analysis purposes, it is useful to have simple access to the iPEPS tensors. In Listing 3, we show how to access the iPEPS tensors associated with the lattice sites $(x, y)$. We use the same clockwise convention as depicted in Fig. 1(b) for indexing the set of boundary tensors by $k = 0, 1, 2, 3$.

```
1 # Access individual A, C, or E tensors for lattice site (x,y)
2 A = ipeps[(x,y)]['A']
3 C = ipeps[(x,y)]['C'][k]
4 E = ipeps[(x,y)]['E'][k]
```

Listing 3: Accessing tensors from an iPEPS instance.

## 4.3 Custom Model Specification

Since models of research interest are very diverse and constantly expanding, it is useful to have fine control over the model and observable specifications. With this in mind, we allow the user to define a custom model class, with methods to define Hamiltonian terms and observables. In Listing 4, we show how a custom model class can be constructed and made available to the iPEPS instance for the usual workflow involving imaginary-time evolution and measurements. We use the Model abstract class defined in the model module of Ace-TN to allow a high level of flexibility in custom model specification. A similar design approach has been shown to be useful recently in the context of model-based quantum chemistry calculations [58].

### 4.3.1 Defining a Custom Model

To define a custom model, the Model class is imported from the model module, a custom model class is created inheriting Model, and the superclass constructor is called. As an example, we consider the quantum compass model Hamiltonian,

$$H = -\sum_{\vec{r}} \left( J_x S_{\vec{r}}^x S_{\vec{r}+\vec{e}_x}^x + J_z S_{\vec{r}}^z S_{\vec{r}+\vec{e}_z}^z \right) \tag{7}$$

which has served as a useful model to explore quantum phase transitions [59, 60] and subextensive degeneracy splitting [60] with iPEPS. The Hamiltonian terms are specified for every bond in the two_site_hamiltonian method of the CompassModel class in Listing 4. We use

the `PauliMatrix` class, which we have included as part of `Ace-TN`, to simplify the construction of spin-based Hamiltonians with multiplication operations overloaded by the Kronecker product. Although, any matrix consistent with the specified physical dimensions can be returned by `two_site_hamiltonian`. We use the `Model` method bond_direction which maps the bond to a direction in {"+x", "-x", "+y", "-y"} of the square lattice to help with specifying bond-dependent Hamiltonians.

```python
from acetn.model import Model
from acetn.model.pauli_matrix import pauli_matrices

class CompassModel(Model):
    def __init__(self, config):
        super().__init__(config)

    def two_site_hamiltonian(self, bond):
        jx = self.params.get("jx")
        jz = self.params.get("jz")
        X,Y,Z,I = pauli_matrices(self.dtype, self.device)
        if self.bond_direction(bond) in ["+x","-x"]:
            return -0.25*jx*X*X
        elif self.bond_direction(bond) in ["+y","-y"]:
            return -0.25*jz*Z*Z
```

Listing 4: Example custom model specification. The hamiltonian terms defining bond interactions for the quantum compass model is defined in `two_site_hamiltonian` and order parameter is defined in `two_site_observables`.

### 4.3.2 Defining Observables

The model class defined for the CompassModel in the previous subsection can now be used for ground-state determination via imaginary-time evolution. However, it is also important to define additional observables to measure using the iPEPS.

An order parameter characterizing the orientation of spins along bond directions can be defined as,

$$\phi = \sum_{\vec{r}} \left\langle \sigma^x_{\vec{r}} \sigma^x_{\vec{r}+\vec{e}_x} - \sigma^z_{\vec{r}} \sigma^z_{\vec{r}+\vec{e}_z} \right\rangle \tag{8}$$

which is negative when spins are oriented along the $\pm z$ directions and positive when spins are oriented along the $\pm x$ directions. For the quantum compass model, this occurs for $J_z > J_x$ and $J_x > J_z$ respectively. We can define the measured operator in the `two_site_observables` method of the custom model class as shown in Listing 5.

```python
def two_site_observables(self, bond):
    observables = {}
    X,Y,Z,I = pauli_matrices(self.dtype, self.device)
    if self.bond_direction(bond) in ["+x","-x"]:
        observables["phi"] = X*X
    elif self.bond_direction(bond) in ["+y","-y"]:
        observables["phi"] = -Z*Z
    return observables
```

Listing 5: Definition of the two_site_observables method of the CompassModel class used to measure the order parameter $\phi$ defined in Eq.(8).

### 4.3.3 Example Parameter Sweep Calculation

A first-order phase transition occurs at the point $J_x = J_z$, which can be seen by adiabatically sweeping the model parameters while maintaining a converged iPEPS. In Listing 6, we show

how this can be achieved. The compass model parameters, $J_x$ and $J_z$ are parameterized by $\theta$ such that $J_x = \cos\theta$ and $J_z = \sin\theta$. The system is then evolved using initial $\theta = 0$ or $\theta = \pi/2$, to initialize a parameter sweep right or left, respectively. After a sufficient number of imaginary-time evolution steps, new model parameters are set by increasing or decreasing $\theta$ by a small amount and set using the set_model_parameters method. After evolution, measurements are obtained by calling the measure method, which returns a dictionary of all measured values averaged over all bonds or sites. The measurements are then saved to a file in comma-separated format using standard python routines.

```python
from acetn.ipeps import Ipeps
from acetn.model import Model
from acetn.model.pauli_matrix import pauli_matrices
import numpy as np
import csv

class CompassModel(Model):
    def __init__(self, config):
        super().__init__(config)

    def two_site_observables(self, bond):
        observables = {}
        X,Y,Z,I = pauli_matrices(self.dtype, self.device)
        if self.bond_direction(bond) in ["+x","-x"]:
            observables["phi"] = X*X
        elif self.bond_direction(bond) in ["+y","-y"]:
            observables["phi"] = -Z*Z
        return observables

    def two_site_hamiltonian(self, bond):
        jx = self.params.get("jx")
        jz = self.params.get("jz")
        X,Y,Z,I = pauli_matrices(self.dtype, self.device)
        if self.bond_direction(bond) in ["+x","-x"]:
            return -0.25*jx*X*X
        elif self.bond_direction(bond) in ["+y","-y"]:
            return -0.25*jz*Z*Z

# function for appending measurement results to file
def append_measurements_to_file(theta, measurements, filename):
    with open(filename, "a", newline="") as file:
        row = [theta,]+[float(val) for val in measurements.values()]
        csv.writer(file).writerow(row)

# initialize an iPEPS
config = {
  "dtype": "float64",
  "device": "cpu",
  "TN":{
    "nx": 2,
    "ny": 2,
    "dims": {"phys": 2, "bond": 2, "chi": 20}
  },
}
ipeps = Ipeps(config)

# set the model
ipeps.set_model(CompassModel, params={"jz":1.0,"jx":1.0})

# sweep parameter range from theta=0 to theta=pi/2
for si in np.arange(0, 1.0, 0.025):
    theta = si*np.pi/2
```

```
53    jx = np.cos(theta)
54    jz = np.sin(theta)
55    ipeps.set_model_params(jx=jx, jz=jz)
56    ipeps.evolve(dtau=0.01, steps=500)
57    ipeps.evolve(dtau=0.001, steps=100)
58    measurements = ipeps.measure()
59    append_measurements_to_file(theta, measurements, "results.dat")
```

Listing 6: Complete script used for calculating and saving results of an iPEPS ground-state calculation obtained from a parameter sweep of the quantum compass model.

The result of the parameter sweep is plotted in Fig. 9. It can be seen that hysteresis occurs as $\theta$ is swept across the transition point $\theta = \pi/4$. Slightly beyond the transition point, the state is adiabatically evolved into a metastable state until eventually becoming unstable and switching to the ground state. This shows how first-order critical points may be determined by imaginary-time evolution of iPEPS and also highlights the importance of choosing a good initial condition or sufficiently large initial imaginary-time step for parameter ranges near the vicinity of a first-order transition.

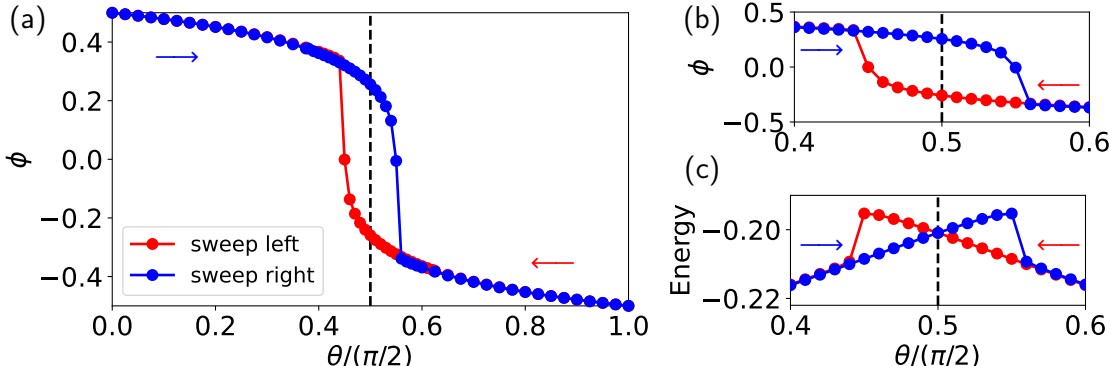

Figure 9: Results of parameter sweep for the compass model. (a) Calculated order parameter $\phi$ which changes discontinuously in the ground state at $\theta = \pi/4$. Sweeping adiabatically from the left ($\theta = 0$) and right ($\theta = 1$) produces the hysteresis effect near $\theta = \pi/4$. (b) Zoom of (a) near the first-order transition. (c) The calculated energy for the iPEPS obtained from left and right sweeps near the transition, the lowest of which corresponds to the ground-state iPEPS. Blue and red arrows indicate the direction of the parameter sweep taken to obtain blue and red data respectively.

### 4.3.4 Setting an Initial condition

By default, the iPEPS will be initialized in a uniform polarized state. This may not be ideal for some systems with ground-state that is expected to differ significantly from the polarized state since this will incur additional evolution steps or possibly result in convergence to a meta-stable state. To set a custom initial condition, there are two options; (1) converge the iPEPS to the ground state of a different model or (2) initialize the system in a product state different than the polarized state. Option (1) is useful when studying first-order phase transitions. For parameter regimes where the stability of the ground state is in question, one may converge to a ground-state near the transition point and adiabatically evolve the system while slowly increasing the model parameters. For option (2), we provide the method, `initial_site_state`, of the Model class.

In Listing 7, we show how an iPEPS can be initialized in a Néel-ordered product state. We use this method for our preset Heisenberg model by default, given that it is much closer to the

ground state than the uniform polarized state.

```
1  def initial_site_state(self, site):
2      xi,yi = site
3      return [1.,0.] if (xi+yi)%2 == 0 else [0.,1.]
```

Listing 7: Example initial condition specification. The initial state is set to the spin-1/2 Néel-ordered product state.

## 4.4 Lattice Coarse Graining

### 4.4.1 Example: Defining Models for the Kagome and Honeycomb Lattices

The CTMRG scheme is most naturally applied to square-lattice systems. However, other infinite 2D systems can be approximated by CTMRG by coarse graining lattice sites, such that the lattice connecting coarse-grained sites is a square lattice. This procedure is easily demonstrated by, but not limited to, the following examples. We show how the coarse-graining may be defined for two common systems: the kagome and honeycomb lattices. For the kagome system, three neighbouring sites may be grouped together as depicted in Fig. 10(a), and one iPEPS site tensor is assigned to the three sites. Similarly, honeycomb-lattice systems can be simulated with CTMRG by grouping two neighbouring sites, as depicted in Fig. 10(b).

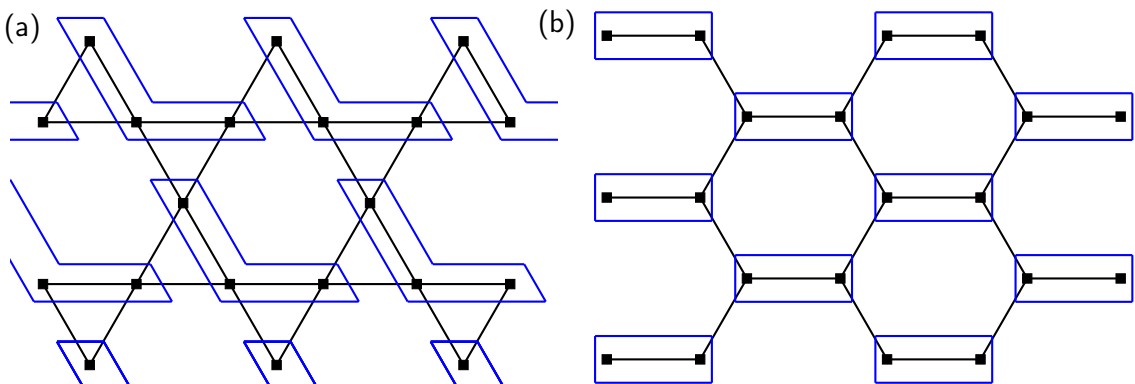

Figure 10: Two examples of common systems, (a) kagome and (b) honeycomb, that can be simulated as an iPEPS by coarse graining. In each case, the black lines represent bond interactions and blue lines surround coarse-grained sites. The coarse-grained sites form a square lattice, and an iPEPS site tensor is assigned to each group of coarse-grained sites.

We show how to define a transverse-field Ising model (TFIM),

$$H = -J_z \sum_{\langle ij \rangle} S_i^z S_j^z - h_x \sum_i S_i^x \tag{9}$$

for the kagome lattice in Listing 8 and honeycomb lattice in Listing 9.

```
1  from acetn.model.pauli_matrix import pauli_matrices
2  from acetn.model import Model
3
4  class KagomeIsingModel(Model):
5      def __init__(self, config):
6          super().__init__(config)
7
8      def one_site_observables(self, site):
9          X,Y,Z,I = pauli_matrices(self.dtype, self.device)
10         mx = (X*I*I + I*X*I + I*I*X)/3
```

```
11          mz = (Z*I*I + I*Z*I + I*I*Z)/3
12          return {"mx": mx, "mz": mz}
13
14      def one_site_hamiltonian(self, site):
15          jz = self.params.get("jz")
16          hx = self.params.get("hx")
17          X,Y,Z,I = pauli_matrices(self.dtype, self.device)
18          return -jz*(I*Z*Z + Z*Z*I) \
19                 -hx*(X*I*I + I*X*I + I*I*X)
20
21      def two_site_hamiltonian(self, bond):
22          jz = self.params.get("jz")
23          X,Y,Z,I = pauli_matrices(self.dtype, self.device)
24          match self.bond_direction(bond):
25              case "-x":
26                  return -jz*(Z*I*I)*(I*I*Z) -jz*(I*Z*I)*(I*I*Z)
27              case "+x":
28                  return -jz*(I*I*Z)*(Z*I*I) -jz*(I*I*Z)*(I*Z*I)
29              case "-y":
30                  return -jz*(I*Z*I)*(Z*I*I) -jz*(I*I*Z)*(Z*I*I)
31              case "+y":
32                  return -jz*(Z*I*I)*(I*Z*I) -jz*(Z*I*I)*(I*I*Z)
```

Listing 8: Custom model class specification corresponding to the kagome-lattice transverse field Ising model.

```
1 from acetn.model.pauli_matrix import pauli_matrices
2 from acetn.model import Model
3
4 class HoneycombIsingModel(Model):
5     def __init__(self, config):
6         super().__init__(config)
7
8     def one_site_observables(self, site):
9         X,Y,Z,I = pauli_matrices(self.dtype, self.device)
10        mx = (X*I + I*X)/2
11        mz = (Z*I + I*Z)/2
12        return {"mx": mx, "mz": mz}
13
14    def one_site_hamiltonian(self, site):
15        jz = self.params.get("jz")
16        hx = self.params.get("hx")
17        X,Y,Z,I = pauli_matrices(self.dtype, self.device)
18        return -jz*(Z*Z) - hx*(X*I + I*X)
19
20    def two_site_hamiltonian(self, bond):
21        jz = self.params.get("jz")
22        X,Y,Z,I = pauli_matrices(self.dtype, self.device)
23        match self.bond_direction(bond):
24            case "+x":
25                return -jz*(I*Z)*(Z*I)
26            case "-x":
27                return -jz*(Z*I)*(I*Z)
28            case "+y":
29                return -jz*(Z*I)*(I*Z)
30            case "-y":
31                return -jz*(I*Z)*(Z*I)
```

Listing 9: Custom model class specification corresponding to the honeycomb-lattice transverse field Ising model.

### 4.4.2 Setting up and Running an iPEPS Calculation

After the model classes are defined, a parameter sweep script can be set up in the same way as in Section 4.3. We show a full code example that can be used to calculate the ground-state iPEPS, measure one-site observables, and save the results to a file called `results.dat` for the `KagomeIsingModel`, and `HoneycombIsingModel` classes in Listing 10.

```python
from acetn.ipeps import Ipeps
import numpy as np
import csv

# function for appending measurement results to file
def append_measurements_to_file(hx, measurements, filename):
    with open(filename, "a", newline="") as file:
        row = [hx,] + [float(val) for val in measurements.values()]
        csv.writer(file).writerow(row)

# initialize an iPEPS
dims = {
    "phys": 8, # use 8 for kagome or 4 for honeycomb
    "bond": 3,
    "chi": 30
}
config = {"TN":{"nx":2, "ny":2, "dims":dims}}
ipeps = Ipeps(config)

# set the iPEPS model (either KagomeIsingModel or HoneycombIsingModel)
ipeps.set_model(KagomeIsingModel, params={"jz":0.25, "hx":0})

# sweep parameter range from hx=0 to hx=2
hx_min = 0
hx_max = 2.0
hx_step = 0.1
for hx in np.arange(hx_min, hx_max, hx_step):
    ipeps.set_model_params(hx=hx/2.)
    ipeps.evolve(dtau=0.01, steps=1000)
    ipeps.evolve(dtau=0.001, steps=200)
    measurements = ipeps.measure()
    append_measurements_to_file(hx, measurements, "results.dat")
```

Listing 10: Full code example used for calculating and saving results from a parameter sweep of the transverse field for the kagome and honeycomb TFIM.

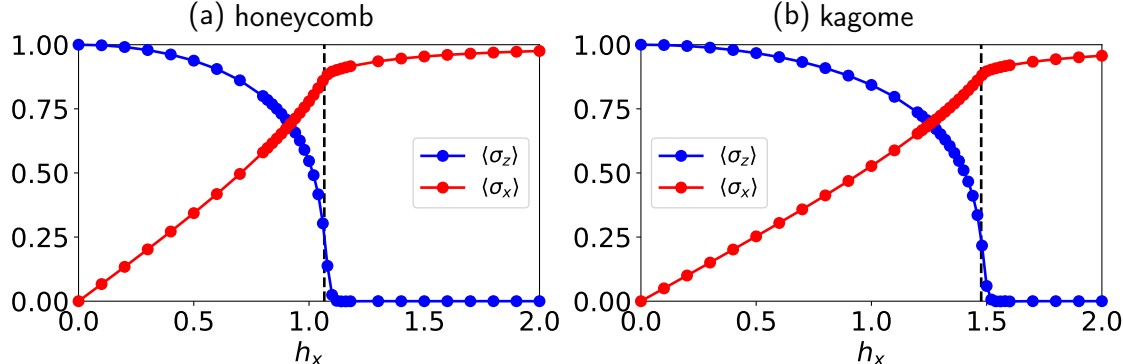

Figure 11: Average on-site magnetizations measured for a range of transverse-fields $h_x$ of the TFIM Hamiltonian defined on the (a) honeycomb lattice and (b) kagome lattice. Dashed lines are plotted using the critical field values from Ref. [61].

The one-site observables defined in Listings 8 & 9 are measured from the converged iPEPS obtained using the script defined in Listing 10 and the results are plotted in Fig. 11. A second-order phase transition can be seen in each system, occuring at critical fields consistent with Monte Carlo estimates [61].

## 5  Conclusion

We have implemented the core algorithms used in the ground-state calculation of an iPEPS including CTMRG and imaginary-time evolution in a GPU-enabled framework. We demonstrate that the CTMRG algorithm, which is the bottleneck in iPEPS calculations, may be greatly accelerated using a single GPU when compared to multithreaded CPU execution. We also propose a simple parallelization scheme capable of attaining good parallel efficiency on a single-node multi-GPU system. Overall, we have shown that a speed-up factor of around 50-60 may be achieved on a single-node multi-GPU system (4×A100) when compared to 20-core CPU execution, or approximately 1000× speedup when compared to single core execution, for typical ground-state calculations involving large bond dimensions. This may be expected since given the most costly operation, the directional move, is embarassingly parallel. While for large $D$ the CPU and GPU execute the $O(D^{12})$ contractions at near peak performance, with the GPU that we use having around 16× higher peak double-precision performance. We have shown that a speed up of around 3.5× may be attained in the directional move by using 4 GPUs, so that an overall speedup of around 3.5×16=56 is reasonable. This implies that some calculations requiring a week of runtime on a single CPU may be completed within around 3 hours using our approach. iPEPS calculations are additionally expected to benefit directly from GPU hardware improvements. We expect to see further 3.4× and 4.6× speed-ups by using the newer Hopper or Blackwell GPU architectures respectively, given the peak double-precision tensor-core performances of 67 TFLOPS/s and 90 TFLOPS/s respectively compared to 19.5 TFLOPS/s of the A100 systems that we used for benchmarking. The multi-GPU implementation that we have developed may also play a vital role in enabling calculations requiring prohibitively large unit cells. For example, as is required to resolve spectral functions from time-evolved iPEPS [62].

Our implementation has been made available in the initial release of the `Ace-TN` library. We have aimed to make `Ace-TN` easily accessible and adaptable to various models of research interest. Several examples of custom Hamiltonian and observable specification have been presented. `Ace-TN` is written entirely using the high-level interface to PyTorch for simplicity. Therefore, we expect that plenty of lower-level optimization may accelerate iPEPS calculations further. While we have focused our initial release around the core algorithms, a plethora of useful additional methods can be added. In future releases of `Ace-TN`, we aim to integrate further algorithms and support specialized use cases. For example, alternative tensor updates [19, 63–65] and CTMRG schemes [66, 67], variational optimization [68] including methods based on automatic differentiation [35, 69, 70], swap gates for fermion simulation [52], exploitation of symmetries [17, 71], real-time evolution [72, 73], spectral function calculation [62, 74, 75], and finite-temperature simulation [72, 73]. Our benchmarks on GPU acceleration indicate a strong potential for speed up across a wide range of iPEPS algorithms.

## Acknowledgements

This research was enabled in part by support provided by SHARCNET (sharcnet.ca) and the Digital Research Alliance of Canada (alliancecan.ca).

**Funding information** The authors acknowledge the support of the Natural Sciences and Engineering Research Council of Canada (NSERC) through Discovery Grants No. RGPIN-2017-05759 and RGPIN-2024-06711. A. D. S. R. acknowledges support from the Ontario Graduate Scholarship.

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
