# Peer review of "Ace-TN: GPU-Accelerated Corner-Transfer-Matrix Renormalization of Infinite Projected Entangled-Pair States"

_SciPost Physics Codebases_

## Round 1 · Referee Report · Anonymous (Referee 1) · 2025-5-7

Strengths

1- The manuscript is clearly written, with the main methods described in a well-structured and comprehensible manner. The grammar and formatting appear to be correct throughout.

2- The results presented are scientifically sound and are either consistent with previously reported findings (such as those related to the Heisenberg model or transverse field Ising model on different lattices) or are well justified from a theoretical perspective (e.g., the performance gains achieved on GPUs).

3- Benchmarking tests clearly prove a significant speed-up in iPEPS calculations that can be achieved on GPUs using Ace-TN library.

4- The accompanying documentation and installation instructions are clear, comprehensive, and easy to follow. The provided script.py example runs smoothly and without issues.

Weaknesses

1- While the manuscript provides a detailed explanation of aspects that are generally well known within the community, less attention is given to more specialized topics—such as randomized SVD (RSVD) or the practical challenges of transitioning iPEPS simulations from CPUs to GPUs—which would benefit from a more thorough discussion.

2- The choice of the fast full update (FFU) method for tensor optimization is well justified; however, as the authors themselves acknowledge, variational methods typically yield better ground state approximations and lower energies. The absence of variational approaches in the library therefore represents a notable limitation.

3- Without sufficient comparison to other PyTorch-based libraries, the necessity and added value of this software for the scientific community are difficult to assess.

Report

Overall, the manuscript meets most of the journal’s criteria. It includes comprehensive benchmarking tests, complete documentation with clear instructions for downloading, installation, and running scripts, and demonstrates multiple example use cases. A detailed application of the method is presented for the Heisenberg model, among other examples. However, the results obtained using the Fast Full Update (FFU) method are suboptimal compared to those achievable with a variational update method (see, for example: https://github.com/jurajHasik/j1j2_ipeps_states/tree/main/single-site_pg-C4v-A1/j20.0). Additional examples demonstrate the use of the software for various models and lattice types. The source code appears modular and well-structured, with informative comments that improve readability, and adheres to high-level programming standards.

However, the manuscript lacks a clear explanation of its added value over existing software packages such as peps-torch or yastn. This raises concerns about whether the tool provides sufficient novelty or practical advantage to encourage adoption by newcomers to the field. Consequently, it is also unclear whether the software addresses a demonstrable need within the scientific community.

To clarify these points and strengthen the manuscript, a revision is necessary. A specific list of recommended changes is provided below.

Requested changes

1- A more comprehensive comparison of Ace-TN library with other libraries (e.g. peps-torch, yastn mentioned in the References) that also support the PyTorch backend—and thus can easily leverage multiple GPUs—would be highly valuable. For larger bond dimensions $D$, the computation time is primarily dominated by tensor contractions, which (I thought) are already optimized for GPU execution in all libraries utilizing PyTorch. Given this, what additional performance benefits does this library offer?

2- The manuscript would greatly benefit from a discussion of the challenges involved in transitioning from CPU to GPU computations, including which subroutines in typical PEPS calculations are readily parallelizable and which require special care. In particular, it would be valuable to explain why practitioners should opt for Ace-TN rather than adapting their own codes for GPU execution (using e.g. Cytnx), or how they might benefit from the insights provided in this manuscript.

3- Does the introduction of RSVD significantly reduce computational time compared to truncated SVD methods like Python's svds? The manuscript would greatly benefit from the inclusion of a pseudo-code for RSVD, as this seems not to be a standard practice and would benefit the community. Additionally, it would be valuable to explore the trade-offs of using an approximate singular value decomposition. Specifically, does the CTM require more iterations to converge when non-exact singular values and vectors are used, as opposed to when the exact decomposition is computed? Furthermore, is it possible to leverage a good initial guess, such as $U$ and $V$ from previous CTM iterations, for low-rank matrix approximation?

4- In the introduction, the authors state "where $\chi$ and $D$ are the bond dimensions of the iPEPS tensors with $\chi \propto D^2$". However, it is unclear where this relation originates. For example, resonating valence bond (RVB) states constructed with relatively low bond dimension (e.g. D = 3) may require a significantly larger $\chi$ to properly converge the boundary, depending on the underlying lattice structure (see, e.g., https://arxiv.org/pdf/1608.06003). Similarly, in practical simulations, $\chi$ is often chosen much larger than $D^2$ to achieve convergence of observables (see,e.g. https://github.com/jurajHasik/j1j2_ipeps_states/tree/main/single-site_pg-C4v-A1/j20.0). Therefore, for clarity and general applicability, it would be helpful if the manuscript consistently expressed computational costs in terms of both $\chi$ and $D$, rather than solely in terms of $D$—even if $\chi=D^2$ is assumed in the authors' specific calculations.

5- The title of Fig. 7(d) is potentially misleading, as it does not clearly specify what quantity the 'fraction' refers to, making it difficult for the reader to interpret the plot accurately.

6- On page 12 the authors write that "We also note that the norm tensor construction scales with $\chi$ as $O(\chi^2)$, while the CTMRG projector calculation scales as $O(\chi^3)$[...]". I suggest revisiting this assessment, as the cost of contracting two $\chi \times \chi$ matrices is $O(\chi^3)$, which seems unavoidable in the construction of the norm tensor.

Recommendation

Ask for major revision

---

## Round 1 · Referee Report · Juraj Hasik (Referee 2) · 2025-5-19

Strengths

1- (to the best of my knowledge) first open-source implementation of CTMRG built with out-of-the-box and easy to use multi-GPU support 2- offers (a comprehensive) benchmark of randomized SVD algorithm used in context of CTMRG , at least within imaginary time evolution context

Weaknesses

1- The multi-GPU parallelization strategy is limited to cases of iPEPS with large unit cells. 2- Absence of validation/test cases of CTMRG with RSVD for states having more complicated/extensive correlations 3- All examples use only real-valued tensors. Complex-valued use case, benchmarking full and/or fast-full update is missing.

Report

On surface, Ace-TN offers basic functionality of many existing iPEPS (or more general) TN libraries, packaged as a compact python software: Well-documented and with a good set of examples serving as a starting point for experimentation by potential users.

From my perspective, its core added value rests in directly addressing one of the challenges iPEPS face: How to utilize heterogenous HPC systems, here focusing on single-node multiple-GPU to accelerate single iPEPS simulation. Ace-TN strategy parallelizes construction of projectors in directional CTMRG over different sites per each row/column in the unit cell over multiple GPUs.

This submission focuses on RSVD, a randomized singular value decomposition, as SVD solver within CTMRG and it brings an account of its performance across few models/lattices. However, my understanding is that Ace-TN is more general - one could easily use different SVD solvers (although this is currently not supported out of the box) and still benefit from the same parallelization strategy.

Provided few modifications (suggested below) are done, I believe this submission will fulfill the criteria for publication in SciPost Codebases.

Requested changes

  • States coming out of imaginary time evolution have, in general, short correlations. For example, as reported in the open dataset https://github.com/jurajHasik/j1j2_ipeps_states, already D=5 iPEPS can give energies lower than D=8 fast full-update optimized states (when more precise optimization methods are used). In such cases, does RSVD work well ? Do default values of hyperparameters (oversampling and iterations), as i.e. set in the source code, lead to correct observables or do they need to be adjustment ?

  • Include complex-valued test case. For example a good one might be Phys. Rev. B 91, 224431 (2015), which gives a single-site D=3 iPEPS family, interpolating between purely real (critical) iPEPS and (bulk-gapped) complex-valued one.

The library would benefit from a more open/documented interface - allowing ease of integration into any other (Python) project:

Think of invoking Ace-TNs CTMRG for iPEPS from other source: * The convention for SiteTensor indices: Is there any particular convention for indices of iPEPS site tensors ? If so, where is it documented ? * The same goes for environment tensors (corners and edges)

  • Docs for renormalization module, i.e. https://ace-tn.github.io/ace-tn/source/acetn.renormalization.html#acetn.renormalization.ctmrg.ctmrg should link to a documented CTMRG config

Recommendation

Ask for minor revision

---

## Editorial Decision

awaiting_resubmission